# OPTIMISTIC ADAPTIVE ACCELERATION FOR OPTIMIZATION

## ABSTRACT

This paper considers a new variant of AMSGrad called Optimistic-AMSGrad. AMSGrad (Reddi et al. (2018)) is a popular adaptive gradient based optimization algorithm that is widely used in training deep neural networks. The new variant assumes that mini-batch gradients in consecutive iterations have some underlying structure, which makes the gradients sequentially predictable. By exploiting the predictability and some ideas from Optimistic Online learning, the proposed algorithm can accelerate the convergence and also enjoys a tighter regret bound. We evaluate Optimistic-AMSGrad and AMSGrad in terms of various performance measures (i.e., training loss, testing loss, and classification accuracy on training/testing data), which demonstrate that Optimistic-AMSGrad improves AMSGrad.

## 1 INTRODUCTION

Nowadays deep learning has been very successful in numerous applications, from robotics (e.g., Levine et al. (2017)), computer vision (e.g., He et al. (2016); Goodfellow et al. (2014)), reinforcement learning (e.g., Mnih et al. (2013)), to natural language processing (e.g., Graves et al. (2013)). A common goal in these applications is learning quickly. It becomes a desired goal due to the presence of big data and/or the use of large neural nets. To accelerate the process, there are variety of training algorithms proposed in recent years, such as AMSGRAD (Reddi et al. (2018)), ADAM (Kingma & Ba (2015)), RMSPROP (Tieleman & Hinton (2012)), ADADELTA (Zeiler (2012)), and NADAM (Dozat (2016)), etc.

All the prevalent algorithms for training deep nets mentioned above combine two ideas: the idea of adaptivity from ADAGRAD (Duchi et al. (2011); McMahan & Streeter (2010)) and the idea of momentum from NESTEROV'S METHOD (Nesterov (2004)) or HEAVY BALL method (Polyak (1964)). ADAGRAD is an online learning algorithm that works well compared to the standard online gradient descent when the gradient is sparse. Its update has a notable feature: the effective learning rate is different for each dimension, depending on the magnitude of gradient in each dimension, which might help in exploiting the geometry of data and leading to a better update. On the other hand, NESTEROV'S METHOD or HEAVY BALL Method (Polyak (1964)) is an accelerated optimization algorithm whose update not only depends on the current iterate and current gradient but also depends on the past gradients (i.e., momentum). State-of-the-art algorithms like AMSGRAD (Reddi et al. (2018)) and ADAM (Kingma & Ba (2015)) leverage the ideas to accelerate training neural nets.

In this paper, we propose an algorithm that goes further than the hybrid of the adaptivity and momentum approach. Our algorithm is inspired by OPTIMISTIC ONLINE LEARNING (see e.g. Chiang et al. (2012); Rakhlin & Sridharan (2013a;b); Syrgkanis et al. (2015); Abernethy et al. (2018)). OPTIMISTIC ONLINE LEARNING considers that a good guess of the loss function in each round is available and plays an action by utilizing the guess. By exploiting the guess, algorithms in OPTIMISTIC ONLINE LEARNING can enjoy a smaller regret than the ones without exploiting the guess. We combine the OPTIMISTIC ONLINE LEARNING idea with the adaptivity and the momentum ideas to design a new algorithm — OPTIMISTIC-AMSGRAD. We also provide a theoretical analysis of OPTIMISTIC-AMSGRAD. The proposed algorithm not only adapts to the informative dimensions, exhibits momentum, but also exploits a good guess of the next gradient to facilitate acceleration. We conduct experiments and show that OPTIMISTIC-AMSGRAD improves AMSGRAD in terms of various measures: training loss, testing loss, and classification accuracy on training/testing data over epochs.

## 2 PRELIMINARIES

We begin by providing some background in online learning, as it will be the main tool to design and analyze our proposed algorithm. We follow the notation in the literature of adaptive optimization (Kingma & Ba (2015); Reddi et al. (2018)). For any vector $u, v \in \mathbb{R}^d$, $u/v$ represents element-wise division, $u^2$ represents element-wise square, $\sqrt{u}$ represents element-wise square-root. We denote $g_{1:T}[i]$ as the sum of the $i_{th}$ element of $T$ vectors $g_1, g_2, \ldots, g_T \in \mathbb{R}^d$.

### 2.1 A BRIEF REVIEW OF ONLINE LEARNING AND OPTIMISTIC ONLINE LEARNING

The standard setup of online learning is that, in each round $t$, an online learner selects an action $w_t \in \mathcal{K} \subseteq \mathbb{R}^d$, then the learner observes $\ell_t(\cdot)$ and suffers loss $\ell_t(w_t)$ after the learner commits the action. The goal of the learner is minimizing the regret,

$$Regret_T(\{w_t\}) := \sum_{t=1}^T \ell_t(w_t) - \sum_{t=1}^T \ell_t(w^*),$$

which is the cumulative loss of the learner minus the cumulative loss of some benchmark $w^* \in \mathcal{K}$.

The idea of OPTIMISTIC ONLINE LEARNING (e.g., Chiang et al. (2012); Rakhlin & Sridharan (2013a;b); Syrgkanis et al. (2015); Abernethy et al. (2018)) is as follows. Suppose that, in each round $t$, the learner has a good guess $m_t(\cdot)$ of the loss function $\ell_t(\cdot)$ before playing an action $w_t$. Then, the learner should exploit the guess $m_t(\cdot)$ to choose an action $w_t$ since $m_t(\cdot)$ is close to the true loss function $\ell_t(\cdot)$. [1] For example, Syrgkanis et al. (2015) proposes an optimistic-variant of FOLLOW-THE-REGULARIZED-LEADER (FTRL). FTRL (see e.g. Hazan (2016)) is an online learning algorithm whose update is

$$w_t = \arg\min_{w \in \mathcal{K}} \langle w, L_{t-1} \rangle + \frac{1}{\eta} R(w),$$

where $\eta$ is a parameter, $R(\cdot)$ is a 1-strongly convex function with respect to a norm ($\|\cdot\|$) on the constraint set $\mathcal{K}$, and $L_{t-1} := \sum_{s=1}^{t-1} g_s$ is the cumulative sum of gradient vectors of the loss functions (i.e., $g_s := \nabla \ell_s(w_s)$ ) up to but not including $t$. FTRL has regret at most $O(\sqrt{\sum_{t=1}^T \|g_t\|_*})$. On the other hand, OPTIMISTIC-FTRL (Syrgkanis et al. (2015)) has the update

$$w_t = \arg\min_{w \in \mathcal{K}} \langle w, L_{t-1} + m_t \rangle + \frac{1}{\eta} R(w),$$

where $m_t$ is the learner's guess of the gradient vector $g_t := \nabla \ell_t(w_t)$. Under the assumption that loss functions are convex, the regret of OPTIMISTIC-FTRL is at most $O(\sqrt{\sum_{t=1}^T \|g_t - m_t\|_*})$, which can be much smaller than the regret of FTRL if $m_t$ is close to $g_t$. Consequently, OPTIMISTIC-FTRL can achieve better performance than FTRL. On the other hand, if $m_t$ is far from $g_t$, then the regret of OPTIMISTIC-FTRL would be only a constant factor worse than that of its non-optimistic counterpart.

In Section 4, we will provide a strategy to obtain $m_t$. At the moment, we just would like to use this example of FTRL to emphasize the importance of leveraging a good guess $m_t$ for updating $w_t$, in order to achieve a faster convergence rate (or equivalently, small regret). We will have a similar argument when we compare OPTIMISTIC-AMSGRAD and AMSGRAD.

### 2.2 ADAM AND AMSGRAD

ADAM (Kingma & Ba (2015)) is a popular algorithm for training deep nets. It combines the momentum idea (Polyak (1964)) with the idea of ADAGRAD (Duchi et al. (2011)), which has effective different learning rates for different dimensions. The effective learning rate of ADAGRAD in iteration $t$ for a dimension $j$ is proportional to the inverse of $\sqrt{\Sigma_{s=1}^t g_s[j]^2}$, where $g_s[j]$ is the $j_{th}$ element of the gradient vector $g_s$ in time $s$. This adaptive learning rate might help for accelerating the convergence when the gradient vector is sparse (Duchi et al. (2011)). However, when applying ADAGRAD to train deep nets, it is observed that the learning rate might decay too fast (Kingma & Ba (2015)). Therefore, Kingma & Ba (2015) propose using a moving average of gradients divided by the square root of the second moment of the moving average (element-wise fashion), for updating the model parameter $w$ (i.e., lines 5,6 and 8 of Algorithm 1). Yet, ADAM (Kingma & Ba (2015)) fails at

---

[1] Imagine that if the learner would had been known $\ell_t(\cdot)$ before committing its action, then it would exploit the knowledge to determine its action and consequently minimizes the regret.

---

**Algorithm 1** AMSGRAD (Reddi et al. (2018))

---

1: Required: parameter $\beta_1$, $\beta_2$, and $\eta_t$.
2: Init: $w_1 \in \mathcal{K} \subseteq \mathbb{R}^d$ and $\hat{v}_0 = v_0 = \epsilon 1 \in \mathbb{R}^d$.
3: **for** $t = 1$ to $T$ **do**
4:   Get mini-batch stochastic gradient vector $g_t$ at $w_t$.
5:   $\theta_t = \beta_1 \theta_{t-1} + (1 - \beta_1) g_t$.
6:   $v_t = \beta_2 v_{t-1} + (1 - \beta_2) g_t^2$.
7:   $\hat{v}_t = \max(\hat{v}_{t-1}, v_t)$.
8:   $w_{t+1} = w_t - \eta_t \frac{\theta_t}{\sqrt{\hat{v}_t}}$. (element-wise division)
9: **end for**

---

some online convex optimization problems. AMSGRAD (Reddi et al. (2018)) fixes the issue. The algorithm of AMSGRAD is shown in Algorithm 1. The difference between ADAM and AMSGRAD lies on line 7 of Algorithm 1. ADAM does not have the max operation on line 7 (i.e., $\hat{v}_t = v_t$ for ADAM) while AMSGRAD adds the operation to guarantee a non-increasing learning rate, $\frac{\eta_t}{\sqrt{\hat{v}_t}}$, which helps for the convergence (i.e., average regret $\frac{\text{Regret}_T}{T} \to 0$). For the parameters of AMSGRAD, it is suggested that $\beta_1 = 0.9$ and $\beta_2 = 0.99$.

## 3 OPTIMISTIC-AMSGRAD

---

**Algorithm 2** OPTIMISTIC-AMSGRAD

---

1: Required: parameter $\beta_1$, $\beta_2$, $\epsilon$, and $\eta_t$.
2: Init: $w_1 = w_{-1/2} \in \mathcal{K} \subseteq \mathbb{R}^d$ and $\hat{v}_0 = v_0 = \epsilon 1 \in \mathbb{R}^d$.
3: **for** $t = 1$ to $T$ **do**
4:   Get mini-batch stochastic gradient vector $g_t$ at $w_t$.
5:   $\theta_t = \beta_1 \theta_{t-1} + (1 - \beta_1) g_t$.
6:   $v_t = \beta_2 v_{t-1} + (1 - \beta_2)(g_t - m_t)^2$.
7:   $\hat{v}_t = \max(\hat{v}_{t-1}, v_t)$.
8:   $w_{t+\frac{1}{2}} = \Pi_{\mathcal{K}} \left[ w_{t-\frac{1}{2}} - \eta_t \frac{\theta_t}{\sqrt{\hat{v}_t}} \right]$.
9:   $w_{t+1} = \Pi_{\mathcal{K}} \left[ w_{t+\frac{1}{2}} - \eta_{t+1} \frac{h_{t+1}}{\sqrt{\hat{v}_t}} \right]$, where $h_{t+1} := \beta_1 \theta_{t-1} + (1 - \beta_1) m_{t+1}$
     and $m_{t+1}$ is the guess of $g_{t+1}$.
10: **end for**

---

We propose a new optimization algorithm, OPTIMISTIC-AMSGRAD, shown in Algorithm 2. In each iteration, the learner computes a gradient vector $g_t := \nabla \ell_t(w_t)$ at $w_t$ (line 4), then it maintains an exponential moving average of $\theta_t \in \mathbb{R}^d$ (line 5) and $v_t \in \mathbb{R}^d$ (line 6), which is followed by the max operation to obtain $\hat{v}_t \in \mathbb{R}^d$ (line 7). The learner also updates an auxiliary variable $w_{t+\frac{1}{2}} \in \mathcal{K}$ (line 8). It uses the auxiliary variable to update and commit $w_{t+1}$ (line 9), which exploits the guess $m_{t+1}$ of $g_{t+1}$ to get $w_{t+1}$. As the learner's action set is $\mathcal{K} \subseteq \mathbb{R}^d$, we adopt the notation $\Pi_{\mathcal{K}}[\cdot]$ for the projection to $\mathcal{K}$ if needed.

We see that OPTIMISTIC-AMSGRAD has three properties:

- Adaptive learning rate of each dimension as ADAGRAD (Duchi et al. (2011)). (line 6, line 8 and line 9)

- Exponentially moving average of the past gradients as NESTEROV'S METHOD (Nesterov (2004)) and the HEAVY-BALL method (Polyak (1964)). (line 5)

- Optimistic update that exploits a good guess of the next gradient vector as optimistic online learning algorithms (e.g. Chiang et al. (2012); Rakhlin & Sridharan (2013a;b); Syrgkanis et al. (2015)). (line 9)

The first property helps for acceleration when the gradient has a sparse structure. The second one is from the well-recognized idea of momentum which can also help for acceleration. The last one, perhaps less known outside the ONLINE LEARNING community, can actually lead to acceleration when the prediction of the next gradient is good. This property will be elaborated in the following subsection in which we provide the theoretical analysis of OPTIMISTIC-AMSGRAD.

Observe that the proposed algorithm does not reduce to AMSGRAD when $m_t = 0$. Furthermore, if $\mathcal{K} = \mathbb{R}^d$ (unconstrained case), one might want to combine line 8 and line 9 and get a single line as $w_{t+1} = w_{t-\frac{1}{2}} - \eta_t \frac{\theta_t}{\sqrt{\hat{v}_t}} - \eta_{t+1} \frac{h_{t+1}}{\sqrt{\hat{v}_t}}$. Yet, based on this expression, we see that $w_{t+1}$ is updated from $w_{t-\frac{1}{2}}$ instead of $w_t$. Therefore, while OPTIMISTIC-AMSGRAD looks like just doing an additional update compared to AMSGRAD, the difference of the updates is subtle. In the following analysis, we show that the interleaving actually leads to certain cancellation in the regret bound.

## 3.1 THEORETICAL ANALYSIS OF OPTIMISTIC-AMSGRAD

We provide the regret analysis here. To begin with, let us introduce some notations first. We denote the Mahalanobis norm $\| \cdot \|_H := \sqrt{\langle \cdot, H \cdot \rangle}$ for some PSD matrix $H$. We let $\psi_t(x) := \langle x, \text{diag}\{\hat{v}_t\}^{1/2} x \rangle$ for a PSD matrix $H_t^{1/2} := \text{diag}\{\hat{v}_t\}^{1/2}$, where $\text{diag}\{\hat{v}_t\}$ represents the diagonal matrix whose $i_{th}$ diagonal element is $\hat{v}_t[i]$ in Algorithm 2. We define its corresponding Mahalanobis norm $\| \cdot \|_{\psi_t} := \sqrt{\langle \cdot, \text{diag}\{\hat{v}_t\}^{1/2} \cdot \rangle}$, where we slightly abuse the notation $\psi_t$ to represent the PSD matrix $H_t^{1/2} := \text{diag}\{\hat{v}_t\}^{1/2}$. Consequently, $\psi_t(\cdot)$ is 1-strongly convex with respect to the norm $\| \cdot \|_{\psi_t} := \sqrt{\langle \cdot, \text{diag}\{\hat{v}_t\}^{1/2} \cdot \rangle}$. Namely, $\psi_t(\cdot)$ satisfies $\psi_t(u) \geq \psi_t(v) + \langle \psi_t(v), u - v \rangle + \frac{1}{2}\|u - v\|_{\psi_t}^2$ for any point $u, v$. A consequence of 1-strongly convexity of $\psi_t(\cdot)$ is that $B_{\psi_t}(u, v) \geq \frac{1}{2}\|u - v\|_{\psi_t}^2$, where the Bregman divergence $B_{\psi_t}(u, v)$ is defined as $B_{\psi_t}(u, v) := \psi_t(u) - \psi_t(v) - \langle \psi_t(v), u - v \rangle$ with $\psi_t(\cdot)$ as the distance generating function. We can also define the corresponding dual norm $\| \cdot \|_{\psi_t^*} := \sqrt{\langle \cdot, \text{diag}\{\hat{v}_t\}^{-1/2} \cdot \rangle}$.

We prove the following result regarding to the regret in the convex loss setting. The proof is available in Appendix B. For simplicity, we analyze the case when $\beta_1 = 0$. One might extend our analysis to more general setting $\beta_1 = [0, 1)$.

**Theorem 1.** *Let $\beta_1 = 0$. Assume that $\mathcal{K}$ has bounded diameter $D_\infty$[2]. Suppose that the learner incurs a sequence of convex loss functions $\{\ell_t(\cdot)\}$.* OPTIMISTIC-AMSGRAD *(Algorithm 2) has regret*

$$Regret_T \leq \frac{1}{\eta_{\min}} D_\infty^2 \sum_{i=1}^d \hat{v}_T^{1/2}[i] + \frac{B_{\psi_1}(w^*, w_{1/2})}{\eta_1} + \sum_{t=1}^T \frac{\eta_t}{2} \|g_t - m_t\|_{\psi_{t-1}^*}^2, \tag{1}$$

*where $g_t := \nabla \ell_t(w_t)$ and $\eta_{\min} := \min_t \eta_t$. The result holds for any benchmark $w^* \in \mathcal{K}$ and any step size sequence $\{\eta_t\}$.*

**Corollary 1.** *Suppose that $v_t$ is always monotone increasing (i.e., $\hat{v}_t = v_t, \forall t$). Then,*

$$Regret_T \leq \frac{1}{\eta_{\min}} D_\infty^2 \sum_{i=1}^d \{(1 - \beta_2) \sum_{s=1}^T \beta_2^{T-s} (g_s[i] - m_s[i])^2\}^{1/2}$$
$$+ \frac{B_{\psi_1}(w^*, w_{1/2})}{\eta_1} + \sum_{t=1}^T \frac{\eta_t}{2} \|g_t - m_t\|_{\psi_{t-1}^*}^2. \tag{2}$$

We should compare the bound of (2)[3] with that of AMSGRAD (Reddi et al. (2018)), which is

$$Regret_T \leq \frac{\sqrt{T}}{2\eta(1-\beta_1)} D_\infty^2 \sum_{i=1}^d \hat{v}_T[i]^2 + D_\infty^2 \sum_{t=1}^T \sum_{i=1}^d \frac{\beta_1 \hat{v}_t[i]^{1/2}}{2\eta_t(1-\beta_1)}$$
$$+ \frac{\eta\sqrt{1+\log T}}{(1-\beta_1)^2(1-\gamma)\sqrt{1-\beta_2}} \sum_{i=1}^d \|g_{1:T}[i]\|_2, \tag{3}$$

where the result was obtained by setting the step size $\eta_t = \eta/\sqrt{t}$. Notice that $\hat{v}_t$ in (3) is the one in Algorithm 1 (AMSGRAD). For fair comparison, let us set $\eta_t = \eta/\sqrt{t}$ in (2) so that $\eta_1 = \eta$ and $\eta_{min} = \eta/\sqrt{T}$ and also let us set $\beta_1 = 0$ in (3) so that their parameters have the same values as ours in the analysis. By comparing the first term in (2) and (3), we clearly see that if $g_t$ and $m_t$ are close, the first term in (2) would be smaller than $\frac{\sqrt{T}}{2\eta(1-\beta_1)} D_\infty^2 \sum_{i=1}^d \hat{v}_T[i]^2$ of (3).

---

[2]The boundedness assumption also appears in the previous works (Reddi et al. (2018); Kingma & Ba (2015)). It seems to be necessary in regret analysis. If the boundedness assumption is lifted, then one might construct a scenario such that the benchmark is $w^* = \infty$ and the learner's regret is infinite.

[3]The following conclusion in general holds for (1), when $v_t$ may not be monotone-increasing. For brevity, we only consider the case that $\hat{v}_t = v_t$, as $\hat{v}_T$ has a clean expression in this case.

Now let us switch to the second term in (2) and (3), we see that $\frac{B_{\psi_1}(w^*, w_{1/2})}{\eta_1} \simeq D_\infty$ in (2), while 0 in (3). For the last term in (2), we have

$$
\sum_{t=1}^{T} \frac{\eta_t}{2} \|g_t - m_t\|^2_{\psi^*_{t-1}}
$$

$$
= \sum_{t=1}^{T-1} \frac{\eta_t}{2} \|g_t - m_t\|^2_{\psi^*_{t-1}} + \eta_T \sum_{i=1}^{d} \frac{(g_T[i] - m_T[i])^2}{\sqrt{v_{T-1}[i]}}
$$

$$
= \sum_{t=1}^{T-1} \frac{\eta_t}{2} \|g_t - m_t\|^2_{\psi^*_{t-1}} + \eta \sum_{i=1}^{d} \frac{(g_T[i] - m_T[i])^2}{\sqrt{T\big((1-\beta_2)\sum_{s=1}^{T-1} \beta_2^{T-1-s}(g_s[i] - m_s[i])^2\big)}}
$$

$$
\leq \eta \sum_{i=1}^{d} \sum_{t=1}^{T} \frac{(g_t[i] - m_t[i])^2}{\sqrt{t\big((1-\beta_2)\sum_{s=1}^{t-1} \beta_2^{t-1-s}(g_s[i] - m_s[i])^2\big)}}.
$$

To interpret the bound, let us make a rough approximation such that

$$
\sum_{s=1}^{t-1} \beta_2^{t-1-s}(g_s[i] - m_s[i])^2 \simeq (g_t[i] - m_t[i])^2.
$$

We can then further obtain an upper-bound as

$$
\sum_{t=1}^{T} \frac{\eta_t}{2} \|g_t - m_t\|^2_{\psi^*_{t-1}} \lessapprox \frac{\eta}{\sqrt{1-\beta}} \sum_{i=1}^{d} \sum_{t=1}^{T} \frac{|g_t[i] - m_t[i]|}{\sqrt{t}} \leq \frac{\eta\sqrt{1+\log T}}{\sqrt{1-\beta}} \sum_{i=1}^{d} \|(g-m)_{1:T}[i]\|_2,
$$

where the last inequality is due to Cauchy-Schwarz. The bound means that when $g_t$ and $m_t$ are sufficiently close, the last term in (2) is smaller than that in (3).

To conclude, as the second term in (2) (which is approximately $D_\infty$) is likely to be dominated by the other terms, the proposed algorithm improves AMSGRAD when the good guess $m_t$ is available.

## 4 PREDICTING $m_t$

From the analysis in the previous section, we know that whether OPTIMISTIC-AMSGRAD converges faster than its counterpart depends on how $m_t$ is chosen. In OPTIMISTIC-ONLINE LEARNING, $m_t$ is usually set to $m_t = g_{t-1}$, i.e., using the previous gradient as a guess of the next one. The choice can accelerate the convergence to an equilibrium in some two-player zero-sum games (Rakhlin & Sridharan (2013a;b); Syrgkanis et al. (2015); Daskalakis et al. (2018)), in which each player uses an optimistic online learning algorithm against its opponent.

This paper is, however, about solving optimization problems instead of solving zero-sum games. We propose to use the extrapolation algorithm of (Scieur et al. (2016)). Extrapolation studies estimating the limit of sequence using the last few iterates (Brezinski & Zaglia (2013)). Some classical works include Anderson acceleration (Walker & Ni. (2011)), minimal polynomial extrapolation (Cabay & Jackson (1976)), reduced rank extrapolation (Eddy (1979)). These methods typically assume that the sequence $\{x_t\} \in \mathbb{R}^d$ has a linear relation

$$
x_t = A(x_{t-1} - x^*) + x^*, \tag{4}
$$

and $A \in \mathbb{R}^{d \times d}$ is an unknown, not necessarily symmetric, matrix. The goal is to find the fixed point of $x^*$. Scieur et al. (2016) relaxes the assumption to certain degrees, by assuming that the sequence $\{x_t\} \in \mathbb{R}^d$ satisfies

$$
x_t - x^* = A(x_{t-1} - x^*) + e_t, \tag{5}
$$

where $e_t$ is a second order term satisfying $\|e_t\|_2 = O(\|x_{t-1} - x^*\|_2^2)$ and $A \in \mathbb{R}^{d \times d}$ is an unknown matrix. The extrapolation algorithm we used is shown in Algorithm 3. Some theoretical guarantees regarding the distance between the output and $x^*$ are provided in (Scieur et al. (2016)).

---

**Algorithm 3** REGULARIZED APPROXIMATE MINIMAL POLYNOMIAL EXTRAPOLATION (RMPE) (Scieur et al. (2016))

---

1: **Input:** sequence $\{x_s \in \mathbb{R}^d\}_{s=0}^{s=r}$, parameter $\lambda > 0$.
2: Compute matrix $U = [x_1 - x_0, \ldots, x_r - x_{r-1}] \in \mathbb{R}^{d \times r}$.
3: Obtain $z$ by solving $(U^\top U + \lambda I)z = \mathbf{1}$.
4: Get $c = z/(z^\top \mathbf{1})$.
5: **Output:** $\Sigma_{i=0}^{r-1} c_i x_i$, the approximation of the fixed point $x^*$.

---

For OPTIMISTIC-AMSGRAD, we use Algorithm 3 to get $m_{t+1}$. The following describes the procedure.

- Call Algorithm 3 with input being a sequence of some past $r + 1$ iterates, $\{w_t, w_{t-1}, w_{t-2}, \ldots, w_{t-r}\}$, where $r$ is a parameter.

- Set $\hat{w}_{t+1} := \Sigma_{i=0}^{r-1} c_i w_{t-r+i}$ from the output of Algorithm 3.

- Output $m_{t+1} := \hat{\nabla} f(\hat{w}_{t+1})$.

That is, the latest $r$ iterates are the input to Algorithm 3. The prediction of the gradient $m_{t+1}$ is by computing a mini-batch stochastic gradient at the output, namely at $\hat{w}_{t+1}$, of Algorithm 3.

We would like to emphasize that the choice of algorithm for gradient prediction is surely not unique. We propose to use the recent result among various related works. Indeed, one can use any method that can provide reasonable guess of the gradient in next iteration.

**Remark:** The work (Scieur et al. (2016)) leverages its extrapolation algorithm to post-process the trajectory of gradient descent and obtains a point that is closer to an optimal point. In contrast, we use the extrapolation algorithm on the fly to accelerate the convergence (of OPTIMISTIC-AMSGRAD).

## 5    EXPERIMENTS

**Datasets and neural nets:** The experiments were conducted on CIFAR10 and CIFAR100 datasets, and a noisy variant of MNIST dataset (MNIST-back-Image (Larochelle et al. (2007)) [4]). We train Res-18 (He et al. (2016)) for CIFAR10 and CIFAR100 datasets and a four-layer convolutional neural net [5] for the noisy MNIST dataset.

In all the experiments described in the following of this section, we use the following hyper-parameters:

- Step size $\eta = 0.001$.

- $\beta_1 = 0.9$, $\beta_2 = 0.99$.

- Number of training samples in each batch: $batch\_size = 64$.

- (Optimistic-AMSGrad) Number of previous iterates stored for gradient prediction: $r = 5$ (i.e., $r = 5$ means the latest five iterates are stored for gradient prediction).

As these are classification tasks, we use the cross entropy loss for training the neural nets. For training on CIFAR 10 and CIFAR 100, after getting the guess of the next iterate $\hat{w}_{t+1}$ by the extrapolation method, we construct the guess of the next gradient $m_{t+1}$ by computing a mini-batch of stochastic gradient of the negative log likelihood loss [6] at $\hat{w}_{t+1}$ (instead of the gradient of the cross entropy loss at $_{t+1}$). The slight modification leads to a better performance.

---

[4] MNIST-back-image takes random patches from a black and white as noisy background. The dataset has 12,000 training samples and 50,000 test samples.

[5] Specifically, we use a neural net model defined on the tutorial page `https://github.com/pytorch/examples/blob/master/mnist/main.py`.

[6] Assume a $K$ classification problem. Denote the values on the output layer $\hat{y}_i \in \mathbb{R}^K$ for a sample $i$ with true label $y_i \in [K]$, its negative log likelihood loss is defined as $-\hat{y}_i[y_i]$. See function nll_loss in PyTorch for details.

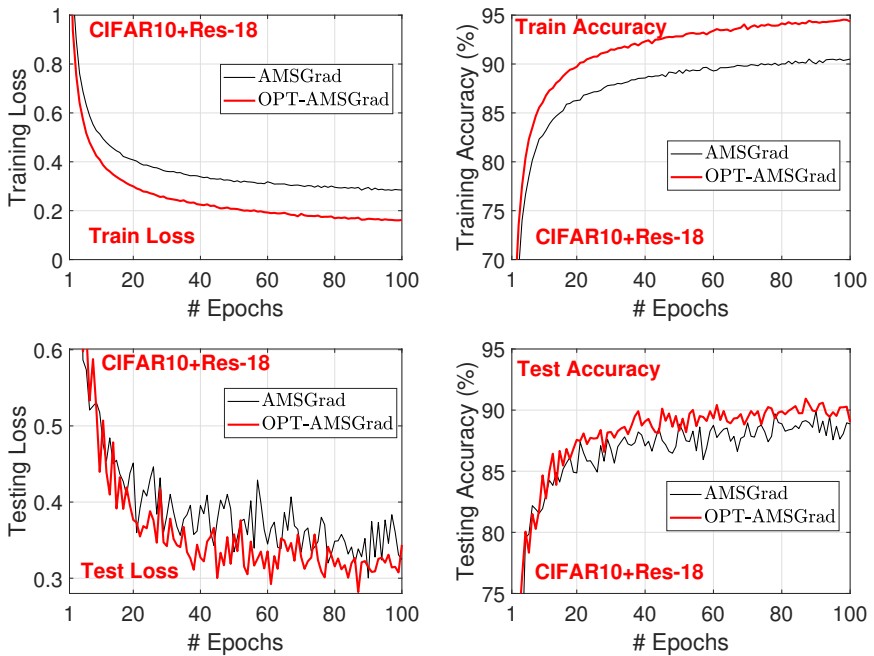

Figure 1: CIFAR 10 + Res-18. We compare OPTIMISTIC-AMSGRAD with AMSGRAD in terms of training (cross-entropy) loss, training accuracy, testing loss, and testing accuracy. All measures are plotted against the numbers of epochs. (One epoch means all training data points are used once). We can see that OPTIMISTIC-AMSGRAD noticeably improves AMSGRAD in all four measures.

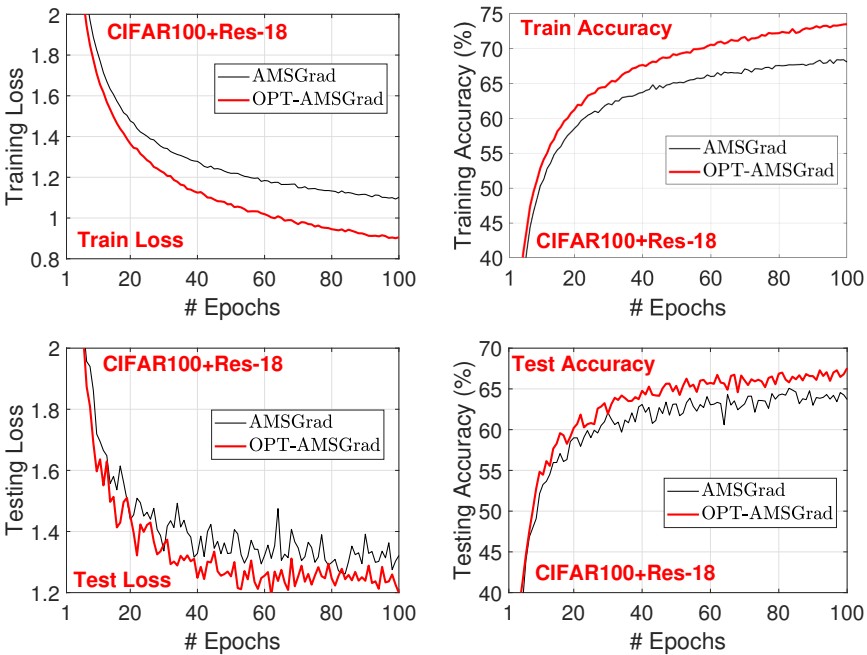

Figure 2: CIFAR 100 + Res-18. We compare OPTIMISTIC-AMSGRAD with AMSGRAD in terms of training (cross-entropy) loss, training accuracy, testing loss, and testing accuracy.

**Results:** Figure 1 shows the result on CIFAR10+Res-18 and Figure 2 shows the result on CIFAR100+Res-18. Figure 3 shows the result on MNIST-back-img dataset.[7] From the results

---

[7]Note that our results on MNIST-back-image actually improved those reported in (Larochelle et al. (2007)), which did not use convolutional nets. The test accuracy is now comparable to that reported in (Li (2010)) which

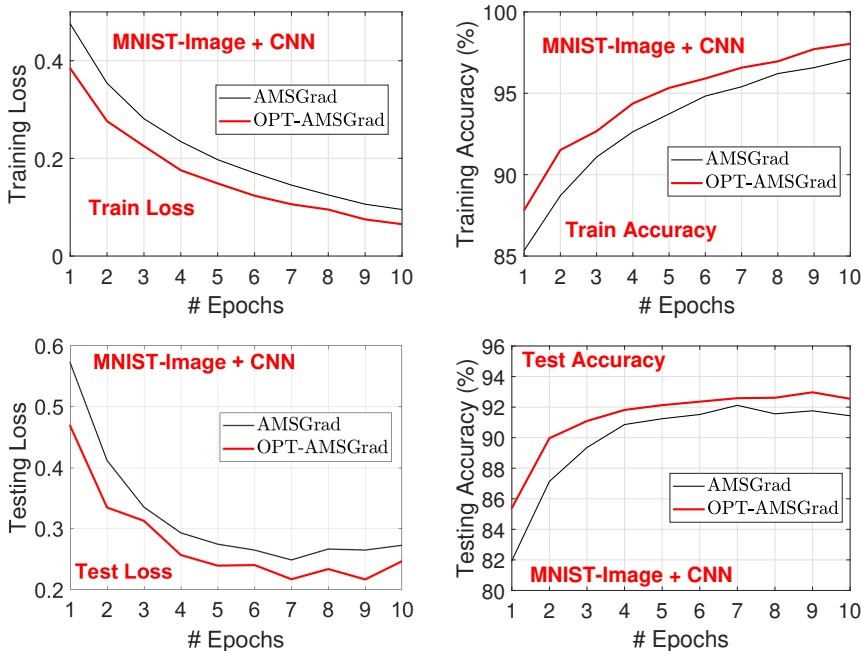

Figure 3: MNIST-back-image + CNN. We compare OPTIMISTIC-AMSGRAD with AMSGRAD in terms of training (cross-entropy) loss, training accuracy, testing loss, and testing accuracy.

shown on the figures, it is clear that OPTIMISTIC-AMSGRAD noticeably improves AMSGRAD in terms of the standard performance measures: training (cross entropy) loss, testing loss, training classification accuracy, and testing classification accuracy. All results are plotted against the number of training epochs. The results also suggest that OPTIMISTIC-AMSGRAD finds a better point that generalize well than AMSGRAD. In Appendix D.2, we report OPTIMISTIC-AMSGRAD with different values of the parameters $r$. We find that the algorithm performance is not sensitive the choice of $r$.

**Comparisons with related works**.    In Appendix A, we provide a comprehensive survey of the related works. There has been a trend in studying adaptive optimization methods from different respects. We compare our contribution and some of the related works, in particular AO-FTRL (Mohri & Yang (2016)) and OPTIMISTIC-ADAM (Daskalakis et al. (2018)). Moreover, in Section D.1, we provide the experimental results for the comparison to a modified version of OPTIMISTIC-ADAM.

## 6    CONCLUSION

We propose OPTIMISTIC-AMSGRAD that combines the ideas of optimistic online learning and AMSGRAD to accelerate optimization. For training deep neural networks, OPTIMISTIC-AMSGRAD significantly improves AMSGRAD in terms of various performance measures in practice (e.g. training loss, testing loss, and classification accuracy on training/testing data). Though we only provide the theoretical analysis in the convex setting, the experiment in non-convex optimization shows some promising results. The results seem to suggest that OPTIMISTIC-AMSGRAD not only minimizes the training loss faster but it can also find a point that generalizes better than the baselines. As the success of OPTIMISTIC-AMSGRAD relies on a good guess of the next gradient, future work includes improving predicting gradients. Exploring the possibility of developing a new way to obtain a better guess of the gradient would be an interesting direction. One possibility is by considering a very recent work of (Dutta et al. (2019)) which proposes a new extrapolation algorithm.

---

developed the second-order tree-split formulation for boosted trees. Also see (Li (2018)) for comparisons with new kernel methods.

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

# A    COMPARISON TO RELATED WORKS

## A.1    COMPARISON TO SOME NON-CONVEX OPTIMIZATION WORKS

Recently, Zaheer et al. (2018); Chen et al. (2019a); Ward et al. (2019); Zhou et al. (2018); Zou & Shen (2018); Li & Orabona. (2019) provide some theoretical analysis of ADAM-type algorithms when applying them to smooth nonconvex optimization problems. For example, Chen et al. (2019a) provide a bound, which is $\min_{t\in[T]} \mathbb{E}[\|\nabla f(w_t)\|^2] = O(\log T/\sqrt{T})$. Yet, this data independent bound does not show an advantage over standard stochastic gradient descent. Similar concerns appear in other papers.

To obtain some adaptive data dependent bound (e.g., bounds like (2) or (3) that are in terms of the gradient norms observed along the trajectory) when applying OPTIMISTIC-AMSGRAD to nonconvex optimization, one can follow the approach of (Agarwal et al. (2019)) or (Chen et al. (2019b)). They provide ways to convert algorithms with adaptive data dependent regret bound for convex loss functions (e.g., ADAGRAD) to the ones that can find an approximate stationary point of non-convex loss functions. Their approaches are modular so that simply using OPTIMISTIC-AMSGRAD as the base algorithm in their methods will immediately lead to a variant of OPTIMISTIC-AMSGRAD that enjoys some guarantee on nonconvex optimization. The variant can outperform the ones instantiated by other ADAM-type algorithms when the gradient prediction $m_t$ is close to $g_t$. We omit the details since this is a straightforward application.

## A.2    COMPARISON TO MOHRI & YANG (2016)

Mohri & Yang (2016) proposes AO-FTRL, which has the update of the form $w_{t+1} = \arg\min_{w\in\mathcal{K}}(\sum_{s=1}^{t} g_s)^\top w + m_{t+1}^\top w + r_{0:t}(w)$, where $r_{0:t}(\cdot)$ is a 1-strongly convex loss function with respect to some norm $\|\cdot\|_{(t)}$ that may be different for different iteration $t$. Data dependent regret bound was provided in the paper, which is $r_{0:T}(w^*) + \sum_{t=1}^{T} \|g_t - m_t\|_{(t)^*}$ for any benchmark $w^* \in \mathcal{K}$. We see that if one selects $r_{0:t}(w) := \langle w, \text{diag}\{\hat{v}_t\}^{1/2}w\rangle$ and $\|\cdot\|_{(t)} := \sqrt{\langle\cdot, \text{diag}\{\hat{v}_t\}^{1/2}\cdot\rangle}$, then the update might be viewed as an optimistic variant of ADAGRAD. However, no experiments was provided in (Mohri & Yang (2016)).

## A.3    COMPARISON TO OPTIMISTIC-ADAM OF (DASKALAKIS ET AL. (2018))

We are aware that Daskalakis et al. (2018) proposed one version of optimistic algorithm for ADAM, which is called OPTIMISTIC-ADAM in their paper. We want to emphasize that the goals are different. OPTIMISTIC-ADAM in their paper is designed to optimize two-player games (e.g., GANs (Goodfellow et al. (2014))), while the proposed algorithm in this paper is designed to accelerate optimization (e.g., solving empirical risk minimization quickly). Daskalakis et al. (2018) focused on training GANs (Goodfellow et al. (2014)). GANs is a two-player zero-sum game. There have been some related works in OPTIMISTIC ONLINE LEARNING like (Chiang et al. (2012); Rakhlin & Sridharan

---

**Algorithm 4** OPTIMISTIC-ADAM (Daskalakis et al. (2018))

---

1: Required: parameter $\beta_1$, $\beta_2$, and $\eta_t$.
2: Init: $w_1 \in \mathcal{K}$.
3: **for** $t = 1$ to $T$ **do**
4:     Get mini-batch stochastic gradient vector $g_t \in \mathbb{R}^d$ at $w_t$.
5:     $\theta_t = \beta_1\theta_{t-1} + (1-\beta_1)g_t$.
6:     $v_t = \beta_2 v_{t-1} + (1-\beta_2)g_t^2$.
7:     $w_{t+1} = \Pi_k[w_t - 2\eta_t\frac{\theta_t}{\sqrt{v_t}} + \eta_t\frac{\theta_{t-1}}{\sqrt{v_{t-1}}}]$.
8: **end for**

---

(2013a;b); Syrgkanis et al. (2015)) showing that if both players use some kinds of OPTIMISTIC-update, then accelerating the convergence to the equilibrium of the game is possible. Daskalakis et al. (2018) were inspired by these related works and showed that OPTIMISTIC-MIRROR-DESCENT can avoid the cycle behavior in a bilinear zero-sum game, which accelerates the convergence. Furthermore, Daskalakis et al. (2018) did not provide theoretical analysis of OPTIMISTIC-ADAM while we give some analysis for the proposed algorithm.

For comparison, we replicate OPTIMISTIC-ADAM in Algorithm 4. OPTIMISTIC-ADAM in Algorithm 4 uses the previous gradient as the guess of the next gradient. Yet, the update cannot be written into the same form as our OPTIMISTIC-AMSGRAD (and vise versa). OPTIMISTIC-AMSGRAD (Algorithm 2) actually uses two interleaving sequences of updates $\{w_t\}_{t=1}^T$, $\{w_{t-\frac{1}{2}}\}_{t=1}^T$. The design and motivation of both algorithms are different.

### A.4 OTHER WORKS ABOUT ADAPTIVE GRADIENT METHODS

There has been a spate of research in improving adaptive gradient methods from different respects. Anil et al. (2019) develop a method to reduce memory overheads in adaptive gradient methods like ADAGRAD and ADAM. Zhou et al. (2019) propose decorrelation between the second moment term $v_t$ and the gradient $g_t$ by temporal shifting to deal with the non-convergence issue of ADAM. Luo et al. (2019) show that an extreme effective learning rate might happen during the execution of ADAM, which can cause the non-convergence. They propose an operation to clip the effective learning rate that avoids the extreme learning rate. Gupta et al. (2018) propose a new adaptive gradient method by designing a preconditioned matrix for the update. Liu et al. (2019) study a heuristic called the learning rate "warm-up" and propose a new variant of ADAM by including a variance rectification term. Becigneul & Ganea (2019) propose a counterpart of ADAM for Riemannian manifolds. Other directions include improving the generalization of adaptive gradient methods (e.g. Loshchilov & Hutter (2019); Chen & Gu (2018); Keskar & Socher. (2017); Luo et al. (2019)), comparing adaptive gradient methods and standard SGD with momentum (e.g. Wilson et al. (2017); Loshchilov & Hutter (2019)), or showing that an adaptive optimization method can escape saddle points (Staib et al. (2019)).

## B PROOF OF THEOREM 1

We provide the regret analysis here. To begin with, let us introduce some notations first. We denote the Mahalanobis norm $\|\cdot\|_H = \sqrt{\langle\cdot, H\cdot\rangle}$ for some PSD matrix $H$. We let $\psi_t(x) := \langle x, \text{diag}\{\hat{v}_t\}^{1/2}x\rangle$ for a PSD matrix $H_t^{1/2} := \text{diag}\{\hat{v}_t\}^{1/2}$, where $\text{diag}\{\hat{v}_t\}$ represents the diagonal matrix whose $i_{th}$ diagonal element is $\hat{v}_t[i]$ in Algorithm 2. We define its corresponding Mahalanobis norm $\|\cdot\|_{\psi_t} := \sqrt{\langle\cdot, \text{diag}\{\hat{v}_t\}^{1/2}\cdot\rangle}$, where we abuse the notation $\psi_t$ to represent the PSD matrix $H_t^{1/2} := \text{diag}\{\hat{v}_t\}^{1/2}$. Consequently, $\psi_t(\cdot)$ is 1-strongly convex with respect to the norm $\|\cdot\|_{\psi_t} := \sqrt{\langle\cdot, \text{diag}\{\hat{v}_t\}^{1/2}\cdot\rangle}$. Namely, $\psi_t(\cdot)$ satisfies $\psi_t(u) \geq \psi_t(v) + \langle\psi_t(v), u-v\rangle + \frac{1}{2}\|u-v\|_{\psi_t}^2$, for any point $u, v$. A consequence of 1-strongly convexity of $\psi_t(\cdot)$ is that $B_{\psi_t}(u,v) \geq \frac{1}{2}\|u-v\|_{\psi_t}^2$, where the Bregman divergence $B_{\psi_t}(u,v)$ is defined as $B_{\psi_t}(u,v) := \psi_t(u) - \psi_t(v) - \langle\psi_t(v), u-v\rangle$ and $\psi_t(\cdot)$ serves as the distance generating function of the Bregman divergence. We can also define the the corresponding dual norm $\|\cdot\|_{\psi_t^*} := \sqrt{\langle\cdot, \text{diag}\{\hat{v}_t\}^{-1/2}\cdot\rangle}$.

*Proof.* [**of Theorem 1**] By regret decomposition, we have that

$$
\begin{aligned}
Regret_T &:= \sum_{t=1}^{T} \ell_t(w_t) - \min_{w \in \mathcal{K}} \sum_{t=1}^{T} \ell_t(w) \\
&\leq \sum_{t=1}^{T} \langle w_t - w^*, \nabla \ell_t(w_t) \rangle \\
&= \sum_{t=1}^{T} \langle w_t - w_{t+\frac{1}{2}}, g_t - m_t \rangle + \langle w_t - w_{t+\frac{1}{2}}, m_t \rangle + \langle w_{t+\frac{1}{2}} - w^*, g_t \rangle,
\end{aligned}
\tag{6}
$$

where we denote $g_t := \nabla \ell_t(w_t)$.

Recall the notation $\psi_t(x)$ and the Bregman divergence $B_{\psi_t}(u, v)$ we defined in the beginning of this section. For $\beta_1 = 0$, we can rewrite the update on line 8 of (Algorithm 2) as

$$
w_{t+\frac{1}{2}} = \arg\min_{w \in \mathcal{K}} \eta_t \langle w, g_t \rangle + B_{\psi_t}(w, w_{t-\frac{1}{2}}),
\tag{7}
$$

and rewrite the update on line 9 of (Algorithm 2) as

$$
w_{t+1} = \arg\min_{w \in \mathcal{K}} \eta_{t+1} \langle w, m_{t+1} \rangle + B_{\psi_t}(w, w_{t+\frac{1}{2}}).
\tag{8}
$$

Now we are going to exploit a useful inequality (which appears in e.g., Tseng (2008)); for any update of the form $\hat{w} = \arg\min_{w \in \mathcal{K}} \langle w, \theta \rangle + B_{\psi}(w, v)$, it holds that

$$
\langle \hat{w} - u, \theta \rangle \leq B_{\psi}(u, v) - B_{\psi}(u, \hat{w}) - B_{\psi}(\hat{w}, v),
\tag{9}
$$

for any $u \in \mathcal{K}$. By using (9) for (8), we have

$$
\langle w_t - w_{t+\frac{1}{2}}, m_t \rangle \leq \frac{1}{\eta_t} \left( B_{\psi_{t-1}}(w_{t+\frac{1}{2}}, w_{t-\frac{1}{2}}) - B_{\psi_{t-1}}(w_{t+\frac{1}{2}}, w_t) - B_{\psi_{t-1}}(w_t, w_{t-\frac{1}{2}}) \right),
\tag{10}
$$

and, by using (9) for (7), we have

$$
\langle w_{t+\frac{1}{2}} - w^*, g_t \rangle \leq \frac{1}{\eta_t} \left( B_{\psi_t}(w^*, w_{t-\frac{1}{2}}) - B_{\psi_t}(w^*, w_{t+\frac{1}{2}}) - B_{\psi_t}(w_{t+\frac{1}{2}}, w_{t-\frac{1}{2}}) \right).
\tag{11}
$$

So, by (6), (10), and (11), we obtain

$$
\begin{aligned}
\text{Regret}_T &\overset{(6)}{\leq} \sum_{t=1}^{T} \langle w_t - w_{t+\frac{1}{2}}, g_t - m_t \rangle + \langle w_t - w_{t+\frac{1}{2}}, m_t \rangle + \langle w_{t+\frac{1}{2}} - w^*, g_t \rangle \\
&\overset{(10),(11)}{\leq} \sum_{t=1}^{T} \| w_t - w_{t+\frac{1}{2}} \|_{\psi_{t-1}} \| g_t - m_t \|_{\psi_{t-1}^*} \\
&+ \frac{1}{\eta_t} \left( B_{\psi_{t-1}}(w_{t+\frac{1}{2}}, w_{t-\frac{1}{2}}) - B_{\psi_{t-1}}(w_{t+\frac{1}{2}}, w_t) - B_{\psi_{t-1}}(w_t, w_{t-\frac{1}{2}}) \right. \\
&+ \left. B_{\psi_t}(w^*, w_{t-\frac{1}{2}}) - B_{\psi_t}(w^*, w_{t+\frac{1}{2}}) - B_{\psi_t}(w_{t+\frac{1}{2}}, w_{t-\frac{1}{2}}) \right),
\end{aligned}
\tag{12}
$$

which is further bounded by

$$
\begin{aligned}
&\overset{(a)}{\leq} \sum_{t=1}^{T} \Big\{ \frac{1}{2\eta_t} \| w_t - w_{t+\frac{1}{2}} \|_{\psi_{t-1}}^2 + \frac{\eta_t}{2} \| g_t - m_t \|_{\psi_{t-1}^*}^2 + \frac{1}{\eta_t} \Big( B_{\psi_{t-1}}(w_{t+\frac{1}{2}}, w_{t-\frac{1}{2}}) - \frac{1}{2} \| w_{t+\frac{1}{2}} - w_t \|_{\psi_{t-1}}^2 \\
&- B_{\psi_{t-1}}(w_t, w_{t-\frac{1}{2}}) + B_{\psi_t}(w^*, w_{t-\frac{1}{2}}) - B_{\psi_t}(w^*, w_{t+\frac{1}{2}}) - B_{\psi_t}(w_{t+\frac{1}{2}}, w_{t-\frac{1}{2}}) \Big) \Big\} \\
&\leq \sum_{t=1}^{T} \Big\{ \frac{\eta_t}{2} \| g_t - m_t \|_{\psi_{t-1}^*} + \frac{1}{\eta_t} \Big( B_{\psi_t}(w^*, w_{t-\frac{1}{2}}) - B_{\psi_t}(w^*, w_{t+\frac{1}{2}}) \\
&+ B_{\psi_{t-1}}(w_{t+\frac{1}{2}}, w_{t-\frac{1}{2}}) - B_{\psi_t}(w_{t+\frac{1}{2}}, w_{t-\frac{1}{2}}) \Big) \Big\},
\end{aligned}
\tag{13}
$$

where $(a)$ is because $\| w_t - w_{t+\frac{1}{2}} \|_{\psi_{t-1}} \| g_t - m_t \|_{\psi_{t-1}^*} = \inf_{\beta > 0} \frac{1}{2\beta} \| w_t - w_{t+\frac{1}{2}} \|_{\psi_{t-1}}^2 + \frac{\beta}{2} \| g_t - m_t \|_{\psi_{t-1}^*}^2$ by Young's inequality and that $\psi_{t-1}(\cdot)$ is 1-strongly convex with respect to $\| \cdot \|_{\psi_{t-1}}$.

To proceed, notice that

$$
\begin{aligned}
B_{\psi_{t+1}}(w^*, w_{t+\frac{1}{2}}) - B_{\psi_t}(w^*, w_{t+\frac{1}{2}}) &= \langle w^* - w_{t+\frac{1}{2}}, \text{diag}(\hat{v}_{t+1}^{1/2} - \hat{v}_t^{1/2})(w^* - w_{t+\frac{1}{2}}) \rangle \\
&\leq (\max_i (w^*[i] - w_{t+\frac{1}{2}}[i])^2) \cdot (\sum_{i=1}^{d} \hat{v}_{t+1}^{1/2}[i] - \hat{v}_t^{1/2}[i])
\end{aligned}
\tag{14}
$$

and

$$B_{\psi_{t-1}}(w_{t+\frac{1}{2}}, w_{t-\frac{1}{2}}) - B_{\psi_t}(w_{t+\frac{1}{2}}, w_{t-\frac{1}{2}})$$
$$= \langle w_{t+\frac{1}{2}} - w_{t-\frac{1}{2}}, \operatorname{diag}(\hat{v}_{t-1}^{1/2} - \hat{v}_t^{1/2})(w_{t+\frac{1}{2}} - w_{t-\frac{1}{2}}) \rangle \le 0, \qquad (15)$$

as the sequence $\{\hat{v}_t\}$ is non-decreasing. Therefore,

$$Regret_T \overset{(13),(14),(15)}{\le} \frac{1}{\eta_{\min}} D_\infty^2 \sum_{i=1}^d \hat{v}_T^{1/2}[i] + \frac{B_{\psi_1}(w^*, w_{1/2})}{\eta_1} + \sum_{t=1}^T \frac{\eta_t}{2} \|g_t - m_t\|_{\psi_{t-1}^*}^2.$$

$\square$

## C  DISCUSSION OF ITERATION COST OF OPTIMISTIC-AMSGRAD

We observe that the iteration cost (i.e., actual running time per iteration) of our implementation of OPTIMISTIC-AMSGRAD is roughly two times larger than the standard AMSGRAD in the empirical minimization task. Here, we report the breakdown analysis for the computational overhead. The overhead mostly comes from the extrapolation step. Specifically, the extrapolation step consists of: (a) The step of constructing the linear system $(U^\top U)$. The cost of this step can be optimized and reduced to $r \times d$, since the matrix $U$ only changes one column at a time. (b) The step of solving the linear system. The cost of this step is $O(r^3)$, which is negligible as the linear system is very small (5-by-5 if $r = 5$). (c) The step that outputs an estimated gradient as a weighted average of previous gradients. The cost of this step is $r \times d$. So, the computational overhead is $2rd + r^3$. Yet, we notice that step (a) and (c) is parallelizable.

**Memory usage:** Our algorithm needs a storage of past $r$ gradients to get an estimated gradient. Though it seems quite demanding compared to the standard AMSGrad, it is relatively cheap compared to Natural gradient method (e.g., Martens & Grosse (2015)), as Natural gradient method needs to store some matrix inverse.

# D   MORE EXPERIMENTS

## D.1   A COMPARISON WITH MODIFIED OPTIMISTIC-ADAM (DASKALAKIS ET AL. (2018))

---

**Algorithm 5** OPTIMISTIC-ADAM+$\hat{v}_t$.

---

1: Required: parameter $\beta_1$, $\beta_2$, and $\eta_t$.
2: Init: $w_1 \in \mathcal{K}$ and $\hat{v}_0 = v_0 = \epsilon 1 \in \mathbb{R}^d$.
3: **for** $t = 1$ to $T$ **do**
4:     Get mini-batch stochastic gradient vector $g_t \in \mathbb{R}^d$ at $w_t$.
5:     $\theta_t = \beta_1 \theta_{t-1} + (1 - \beta_1)g_t$.
6:     $v_t = \beta_2 v_{t-1} + (1 - \beta_2)g_t^2$.
7:     $\hat{v}_t = \max(\hat{v}_{t-1}, v_t)$.
8:     $w_{t+1} = \Pi_k[w_t - 2\eta_t \frac{\theta_t}{\sqrt{\hat{v}_t}} + \eta_t \frac{\theta_{t-1}}{\sqrt{\hat{v}_{t-1}}}]$.
9: **end for**

---

Here we also compare OPTIMISTIC-AMSGRAD with another baseline, which we called OPTIMISTIC-ADAM+$\hat{v}_t$ as shown in Algorithm 5. OPTIMISTIC-ADAM+$\hat{v}_t$ is OPTIMISTIC-ADAM (Algorithm 4) of (Daskalakis et al. (2018)) with the additional max operation $\hat{v}_t = \max(\hat{v}_{t-1}, v_t)$ to guarantee that the weighted second moment is monotone increasing. Figure 4, 5, and 6 show the results. We observe that our method dominates the other two methods.

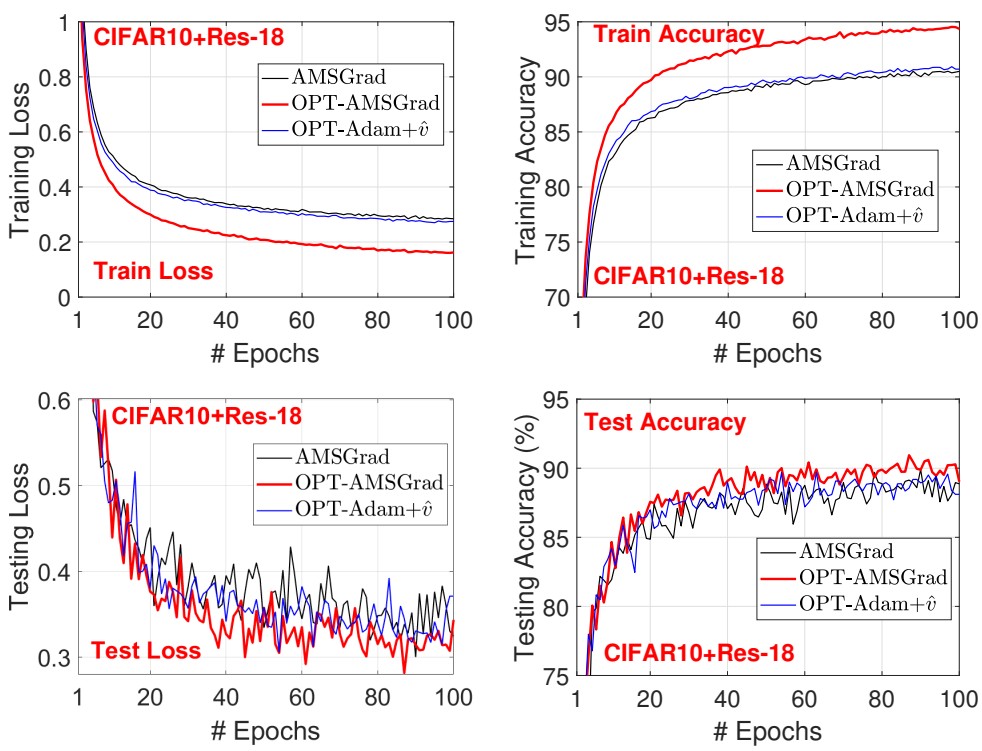

Figure 4: CIFAR 10 + Res-18. We compare three methods: OPTIMISTIC-AMSGRAD, AMSGRAD, and OPTIMISTIC-ADAM+$\hat{v}_t$, in terms of training (cross-entropy) loss, training accuracy, testing loss, and testing accuracy. We observe that OPTIMISTIC-AMSGRAD consistently improves the two baselines.

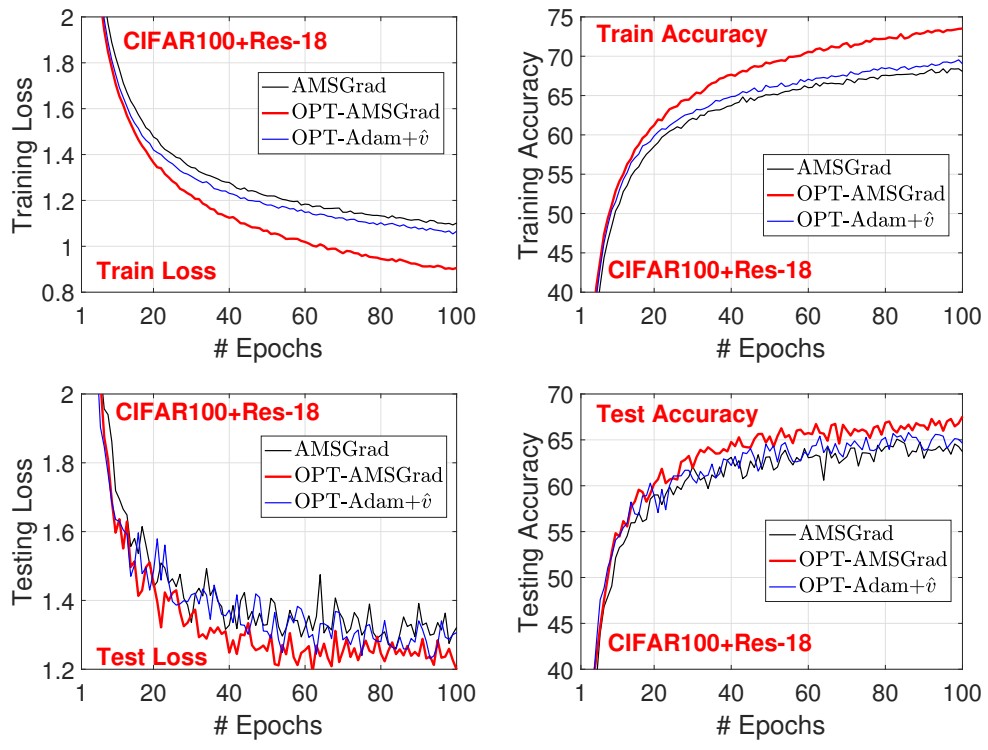

Figure 5: CIFAR 100 + Res-18. We compare three methods: OPTIMISTIC-AMSGRAD, AMSGRAD, and OPTIMISTIC-ADAM+$\hat{v}_t$, in terms of training (cross-entropy) loss, training accuracy, testing loss, and testing accuracy.

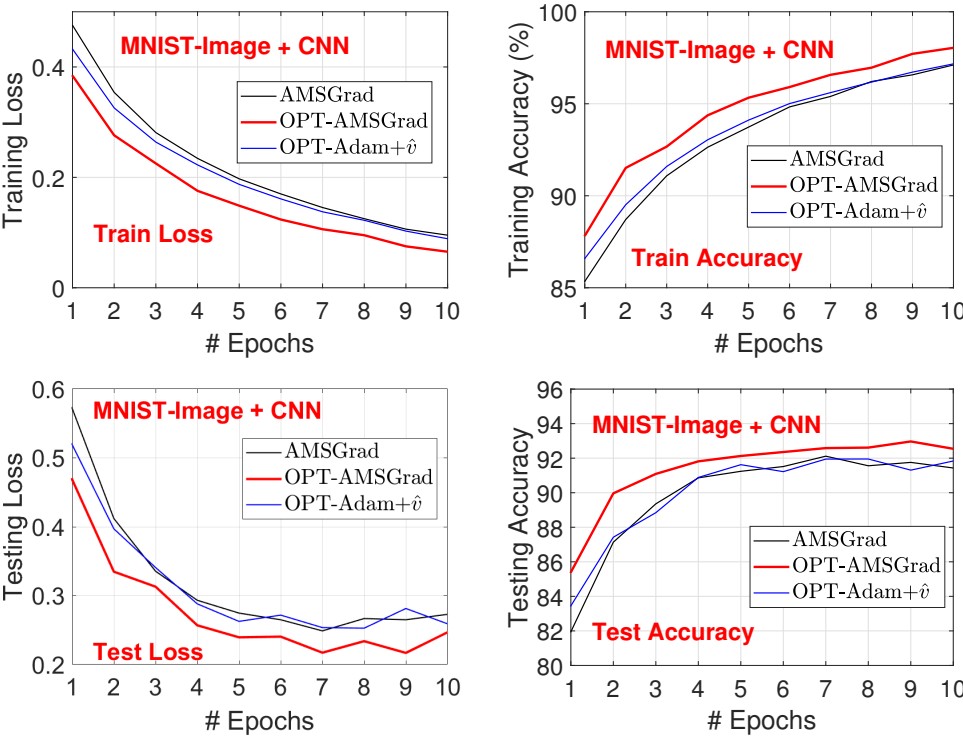

Figure 6: MNIST-back-image noisy dataset + a four-layer convolutional neural network.

## D.2 CHOICE OF DIFFERENT $r$ VALUES

Recall that our proposed algorithm has the parameter $r$ in addition to the step size $\eta$ that governs the use of past information. Figure 7, 8, and 9 compare the performance under different values or $r$, $r = 3, 5, 10$. From the result we see that the choice of $r$ does not have significant impact on learning performance. Taking consideration both quality of gradient prediction and computational issues, it appears that $r = 5$ is a good choice, although the results for $r = 3$ do not differ much.

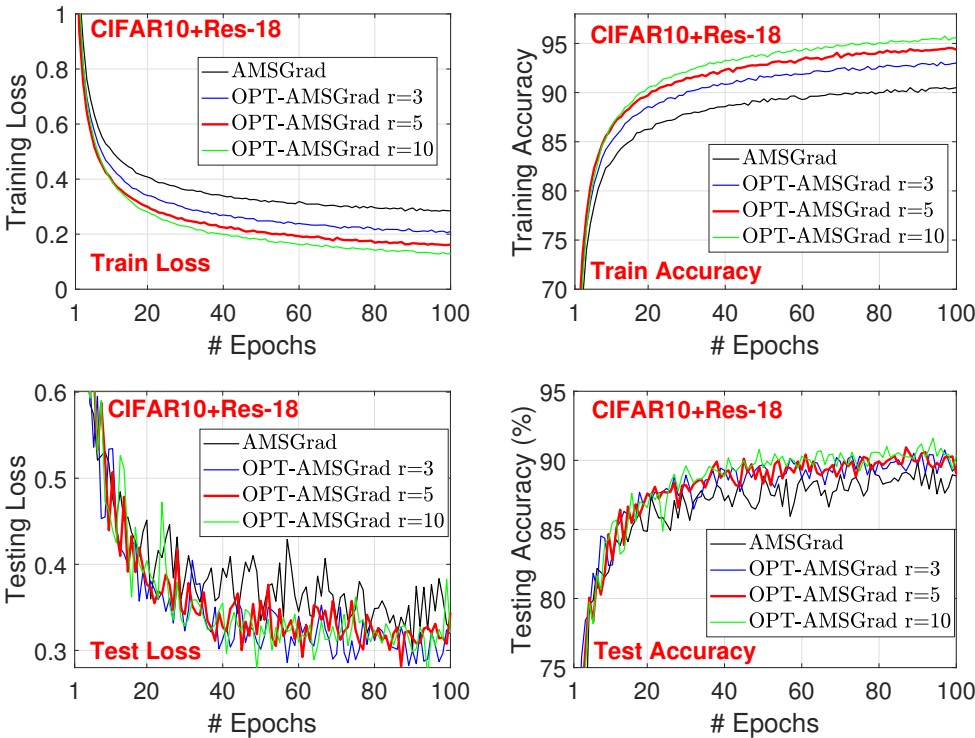

Figure 7: CIFAR 10 + Res-18. We compare OPTIMISTIC-AMSGRAD (for $= 3, 5, 10$) with AMSGRAD in terms of training (cross-entropy) loss, training accuracy, testing loss, and testing accuracy. The choice of $r$ does not have significant impact on learning performance. While it appears that $r = 5$ is a good choice, the results for $r = 3$ do not differ much.

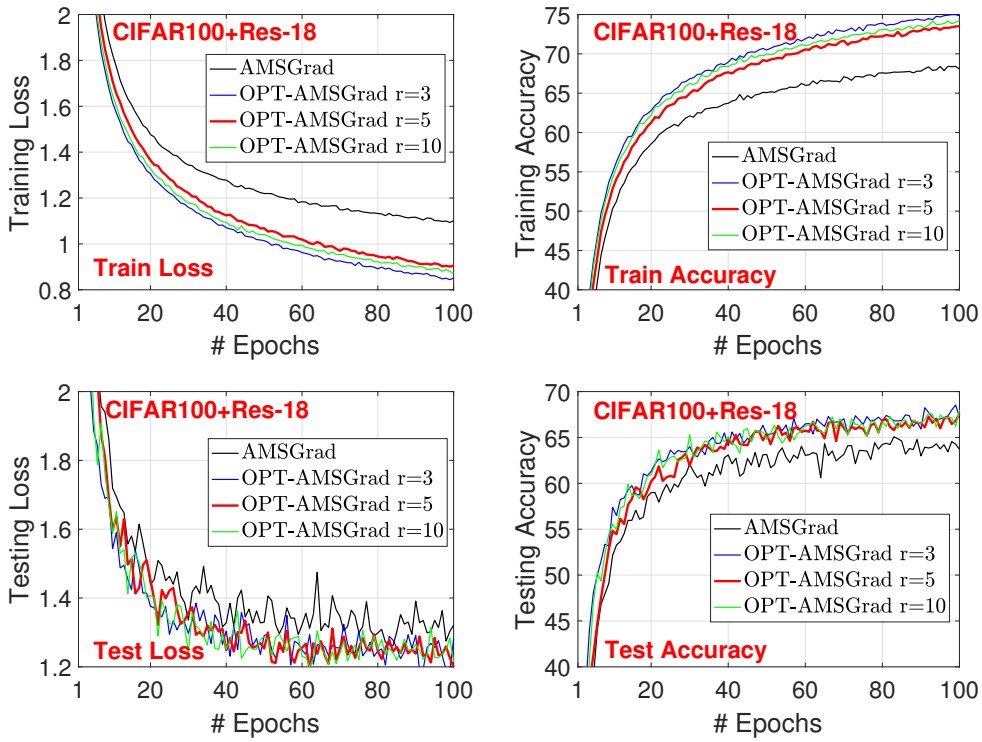

Figure 8: CIFAR 100 + Res-18. We compare OPTIMISTIC-AMSGRAD (for $= 3, 5, 10$) with AMSGRAD in terms of training (cross-entropy) loss, training accuracy, testing loss, and testing accuracy. Again, the choice of $r$ does not affect the results too much.

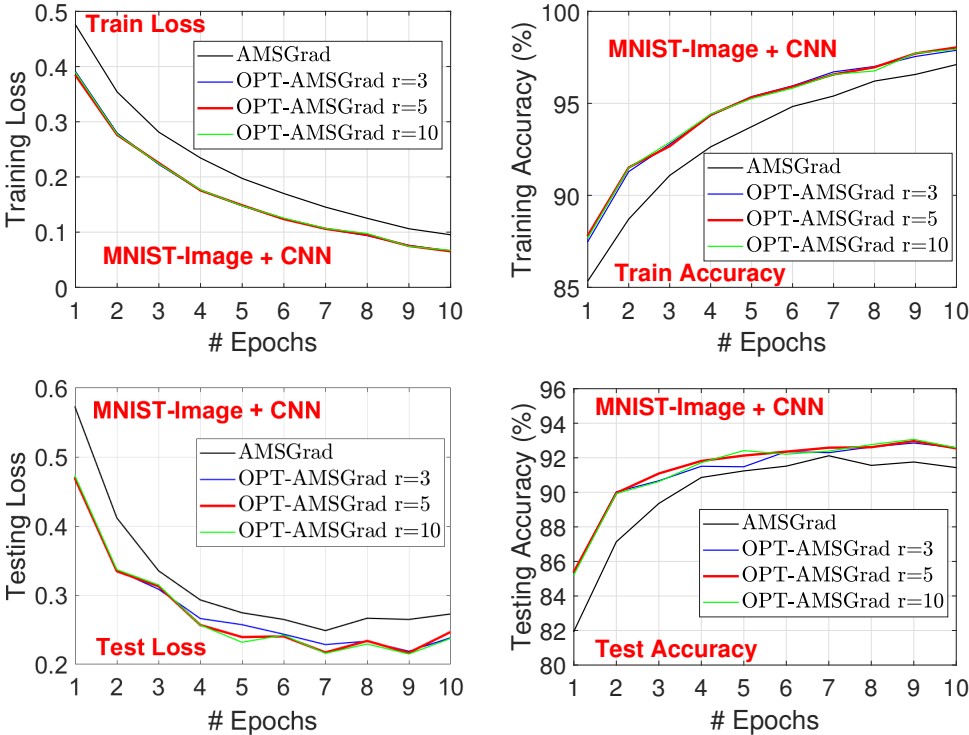

Figure 9: MNIST-back-image noisy dataset + a four-layer convolutional neural network.

