# OpenReview forum: "Optimistic Adaptive Acceleration for Optimization"
_ICLR.cc/2020/Conference — Reject_

### Official Review · AnonReviewer3 · 2019-10-22
**Official Blind Review #3**

**Rating:** 3

**Review:**

Summary:


This work proposed a new variant of AMSGrad called Optimistic-AMSGrad, which makes use of the ideas from Optimistic Online learning. The authors showed that Optimistic-AMSGrad enjoys lower regret compared with AMSgrad in online learning. Experiment results backup their theory.

Pros:

This work proposed a new variant of AMSGrad called Optimistic-AMSGrad. In the paper the authors showed that by predicting the future gradient using m_t, the regret of Optimistic-AMSGrad can be lowered from \sum |g_t| to \sum |g_t - m_t|, which improves AMSGrad directly. The authors also gave a practical way to compute m_t based on history information with the underlying assumption on input x_t. The authors provided detailed experiment results to backup their theory.

Cons:

- There is no discussion about the choice of parameters. From equation 2, Corollary 1, it seems that to set \beta_2 = 1 achieves the best regret, which implies that to keep v_t unchanged achieves the best result. That sounds a bit strange because it suggests that the coordinate correction is useless. I recommend the authors to add some explanation for their corollary here.
- The intuition behind Algorithm 3 should be demonstrated more clear. Right now I do not understand how the correlation between x_t affects the prediction of m_t. The authors should add more explanation in Section 3.
- The experiment results are not well aligned with theoretical results, since the authors considered convex loss in their proof, while the optimization on neural network is a highly non-convex task. I suggest the authors add some simple convex examples to demonstrate the superiority of Optimistic-AMSGrad.


**Experience Assessment:**

I have published one or two papers in this area.

**Review Assessment: Checking Correctness Of Derivations And Theory:**

I carefully checked the derivations and theory.

**Review Assessment: Checking Correctness Of Experiments:**

I carefully checked the experiments.

**Review Assessment: Thoroughness In Paper Reading:**

I read the paper thoroughly.

---

> ### Author Response · Authors · 2019-11-15
> **Response to Reviewer 3**
>
> Thanks for your valuable suggestions.
> 1)	The assumption  $\beta_1=0$ is mainly for the ease of analysis. This assumption is also adopted in
> Manzil Zaheer, Sashank Reddi, Devendra Sachan, Satyen Kale, and Sanjiv Kumar. Adaptive methods for nonconvex optimization, NeurIPS 2018
> 2)	We use the RMPE algorithm (Algorithm 3) as a straightforward application for gradient prediction, so we did not include more detailed explanation on it. We will definitely add some given more space.

---

### Official Review · AnonReviewer2 · 2019-10-23
**Official Blind Review #2**

**Rating:** 3

**Review:**

This paper proposes an online optimization method called Optimistic-AMSGrad, which combines two existing methods: (i) AMSGrad (Reddi et al 2018) and (ii) optimistic online learning where the prediction step is done with the extrapolation algorithm by Scieur et al 2016. The authors do a good job of presenting the method (by introducing the background in proper order), the paper seems self-contained and cites the relevant literature. The regret analysis of the proposed algorithm is provided, where the obtained regret can be smaller than AMSGrad depending on whether or not the guess of the gradient and the gradient are close.

In my opinion the boundedness assumption (footnote 2) is quite important here, and should be mentioned in the main text.

It is not clear how the different ways of accelerations combined in this method interact when the guess is not good. In other words, if the guess is not good this method could be slower then AMSGrad. Moreover, AMSGrad has stability property allowed from the ratio between 1st and 2nd moment estimate. In Optimistic-AMSGrad if m_t is bad, obtaining next w_{t+1} (line 9) would include ratio between outdated/bad 1st mom. estimate and new 2nd-moment estimate. In short, the method’s stability and outperformance might rely on the selection of the algorithm for gradient prediction.

In my understanding, extragradient has clear advantages in games, as if considering simple bilinear examples it is the only method that converges. However, for a single objective, its advantages are not clear to me (after reading the paper). Thus, I think it would be useful if the authors could provide comparison over *wall clock time* as well as long-run comparisons when the compared methods converge (it would be interesting to see if Optimistic-AMSGrad obtains better final train/test accuracy?). In many of the experiments where Opt-AMSGrad outperforms, the accuracy of the baseline still goes up--whereas the latter is computationally cheaper, so it is not clear from the provided results why a practitioner should use this method.
Moreover, the experimental results would be much more convincing if the authors do multiple runs using different seeds and present mean and standard deviation of the methods.
In the context of games, using more computationally demanding optimizers makes sense as training is unstable. In this case, after reading the paper, it is not clear to me what is the problem that the proposed method solves (or its advantages). Indeed, its advantage depends on how good the guess of g_{t+1} is. However, the extra-computation cost to obtain a good guess needs to be justified, or proven empirically that gets better performances faster (wall-clock time), or final ones.

In summary: (i)  the paper is well-presented and provides hyperparameter sensitivity results; (ii) the paper is very interesting, but (imo) it should leave clearer message why one should use this method; (iii) the proposed method has tighter regret, but only in some (data-dependent) cases and combines existing methods, limiting novelty. Hence, given the pros and cons, I am not confident recommending acceptance, and I think improved experimental results (error bars & wall-clock, see above) would make the results more significant.


--- Minor ---
- Alg.2: maybe add input hyperparameter r to optimistic-AMSGrad and a line between lines 8-9 that calls function which obtains the guess m_{t+1} (r should be passed to it). It would make it more clear that your algorithm has parameter r (sec. D.2)
- I could be wrong, but in my opinion, using \theta as first moment est. is slightly confusing, as it is normally used to denote parameters; similarly, the authors could use hat/prime on top of a variable to denote the ‘guess’ of that same variable, making it easier to follow.
- maybe add init of \theta_0 in alg1&2
- if I am correct amsgrad also does bias correction of the initial values of 1st and 2nd moment estimates; if that’s the case it could be useful adding a note that this is omitted for clarity if a reader implements it
- pg2: we just would like -> we would like


**Experience Assessment:**

I have published one or two papers in this area.

**Review Assessment: Checking Correctness Of Derivations And Theory:**

I assessed the sensibility of the derivations and theory.

**Review Assessment: Checking Correctness Of Experiments:**

I assessed the sensibility of the experiments.

**Review Assessment: Thoroughness In Paper Reading:**

I read the paper thoroughly.

---

> ### Author Response · Authors · 2019-11-15
> **Response to Reviewer 2**
>
> Thanks for your valuable feedback and suggestions.
> 1)	Yes, the quality of the guess $m_t$ is very important in OPT-AMSGRAD. As a first attempt, we try to use RMPE and empirically find it effective. More research towards this direction is surely meaningful and interesting.
> 2)	Our results are all averaged over 5 trials. We apologize that we did not mention it in the submission. The benefit of “optimistic + online learning” is the improved sample efficiency, which means that with same number of samples used (e.g. fixed number of epochs), the OPT-AMSGRAD converges faster than the baseline, which could be seen from the experiments. In addition, we find that OPT-AMSGRAD is able to provide better test accuracy as well. The extra cost mainly comes from computing one more gradient in each step. We will provide some more detailed explanation on the computational cost.

---

### Official Review · AnonReviewer1 · 2019-10-23
**Official Blind Review #1**

**Rating:** 3

**Review:**

This paper studies an optimistic variant of AMSGrad algorithm, where an estimate of the future gradient is incorporated into the optimization problem. The main claim is that when we have good enough (distance from the ground truth is small) estimate of the unknown gradient, the proposed algorithm will enjoy lower regret. Theoretical results are provided and experiments are conducted to compare the proposed algorithm with baselines. The idea seems to be not very novel since the optimistic optimization techniques are borrowed directly from the online optimization field, while it is still interesting to see this kind of work and to see its comparison with existing algorithms in experiments. However, the comparison seems to be not fair both in theory and experiments.

In the second paragraph of Section 2.1, you use $m_t$ to denote an estimate of the loss function. But later (In the third equation) you use $m_t$ to denote the guess of gradient vector. The notation is reloaded without any description, which makes the presentation confusing.

In addition, at the end of this paragraph, you mentioned that even when $m_t$ is far away from $g_t$, the regret of an optimistic algorithm is just a constant factor of non-optimistic one. This seems not rigorous since it is true only when the divergence of $m_t$ from $g_t$ is in constant order.

Can you explain why in line 8 of Algorithm 2, you use $w_{t-½}$ to update $w_{t+1}$ instead of $w_{t}$? A discussion about these choices should be added to the description of algorithm.

In the first equation on page 5, the second term on the R.H.S. of the first equation misses a factor of $1/2$. Moreover, the second equation should be inequality.

In the comparison of equation (2) and (3), I think it should be pointed out that when the gradient has a sparse structure, the regret of the optimistic-AMSGrad in (2) seems to be worse than that of the original AMSGrad.

The optimistic algorithm seems to cost more computation in order to estimate the unknown gradient in advance. In the experiment part, you used the last few iterations to estimate the guess of gradient in the next step. But it seems that the comparison with $r=3,5,10$ is not consistent in many plots since I expected a larger $r$ will lead to more accurate estimate.

The experiment results seem to be not convincing. In particular, in Fig. 1,2 and others, the training loss of AMSGrad is far away from zero, which implies that the algorithm is not fully optimized. Therefore, it is hard to draw any meaningful conclusion from the current experiments.

In footnote 1, “had been known” -> “had known”

====after rebuttal
The authors did not provide satisfying response neither submit any revision to address the questions. I will keep my rating.

**Experience Assessment:**

I have published one or two papers in this area.

**Review Assessment: Checking Correctness Of Derivations And Theory:**

I assessed the sensibility of the derivations and theory.

**Review Assessment: Checking Correctness Of Experiments:**

I assessed the sensibility of the experiments.

**Review Assessment: Thoroughness In Paper Reading:**

I read the paper at least twice and used my best judgement in assessing the paper.

---

> ### Author Response · Authors · 2019-11-15
> **Response to Reviewer 1**
>
> Thanks for your valuable suggestions on improving the paper.
> 1)	For Algorithm2, the update is based on the standard optimistic updating strategy using the “half-gradient”.
> 2)	For the experiments, the results of r=3,5,10 not giving a very consistent pattern is partly because that the starting iteration of using the optimistic step is also different (at iteration 3,5,10 respectively, for collecting enough gradients). This brings uncertainty since early start may be either good or bad, depending on different datasets. However, we can see that the test performance is very similar, so we suggest setting r around 5 as a good choice in practice.
>
> The goal of OPTIMISTIC-AMSGRAD is to improve the sample efficiency: using same number of samples (for example, fixed number of epochs), OPT-AMSGRAD can converge much faster than AMSGRAD and achieve better performance. Furthermore, the test accuracy has become stable in Figure 1,2 and 3, which indicates that OPT-AMSGRAD is better than AMSGRAD in testing phase. For the training loss plots, we try to emphasize the acceleration effect of OPT-AMSGRAD.

---

### Decision · Program_Chairs · 2019-12-19

**Decision:**

Reject

**Comment:**

The paper introduces a variant of AMSGrad ("Optimistic-AMSGrad"), which integrates an estimate of the future gradient into the optimization problem. While the method is interesting, reviewers agree that novelty is on the low side. The motivation of the approach should also be clarified. The experimental section should be made stronger; in particular, reporting convincing wall-clock running time advantages is critical for validating the viability of the proposed approach.